# MicroRNAs—The Heart of Post-Myocardial Infarction Remodeling

**DOI:** 10.3390/diagnostics11091675

**Published:** 2021-09-13

**Authors:** Liana Maries, Cătălin Marian, Raluca Sosdean, Flavia Goanta, Ioan Ovidiu Sirbu, Andrei Anghel

**Affiliations:** 1Biochemistry Department, “Victor Babes” University of Medicine and Pharmacy Timisoara, 300041 Timisoara, Romania; maries.liana@umft.ro (L.M.); cmarian@umft.ro (C.M.); ralusosdean@yahoo.com (R.S.); flavia2112@yahoo.com (F.G.); biochim@umft.ro (A.A.); 2Center for Complex Networks Science, “Victor Babes” University of Medicine and Pharmacy Timisoara, 300041 Timisoara, Romania; 3Institute of Cardiovascular and Heart Disease of Timisoara, 300310 Timisoara, Romania

**Keywords:** miRNA, myocardial infarction, heart failure, left ventricular remodeling, biomarker, prognostic

## Abstract

Myocardial infarction (MI) is one of the most frequent cardiac emergencies, with significant potential for mortality. One of the major challenges of the post-MI healing response is that replacement fibrosis could lead to left ventricular remodeling (LVR) and heart failure (HF). This process involves canonical and non-canonical transforming growth factor-beta (TGF-β) signaling pathways translating into an intricate activation of cardiac fibroblasts and disproportionate collagen synthesis. Accumulating evidence has indicated that microRNAs (miRNAs) significantly contribute to the modulation of these signaling pathways. This review summarizes the recent updates regarding the molecular mechanisms underlying the role of the over 30 miRNAs involved in post-MI LVR. In addition, we compare the contradictory roles of several multifunctional miRNAs and highlight their potential use in pressure overload and ischemia-induced fibrosis. Finally, we discuss their attractive role as prognostic biomarkers for HF, highlighting the most relevant human trials involving these miRNAs.

## 1. Introduction

Coronary artery disease (CAD) is the leading cause of death for both women and men in developed as well as in developing countries, acute myocardial infarction (MI) being responsible for most of the CAD mortality [1]. Over the past two decades, MI mortality and morbidity have significantly decreased due to extensive prevention programs and effective revascularization procedures. Furthermore, novel drug therapies and standard of care protocols have been implemented to avoid further complications such as life-threatening arrhythmias and heart failure [2].

Post-MI heart failure is the result of left ventricular remodeling (LVR). LVR refers to the ventricular shape, size, and function changes consequent to cardiomyocyte apoptosis and hypertrophy [3]. Ultimately, fibrosis leads to impairment of the left ventricle’s systolic and diastolic functions [4], underlining the need for LVR early predictive biomarkers and preventive therapies.

Currently, several circulating biomarkers could predict with decent accuracy post-MI cardiovascular events, of which, N-terminal pro-brain natriuretic peptide (NT-pro-BNP) and cardiac troponin (cTn) are the most promising [5]. However, their use is limited, primarily due to their fluctuating levels in the blood and the influence of other comorbidities, such as renal or hepatic failure [6].

MicroRNAs (miRNAs) are a class of small non-coding RNAs that post-transcriptionally regulate gene expression by translational repression or messenger RNA (mRNA) degradation [7]. MiRNAs can, either directly or via other effectors, modulate the remodeling process after MI. The balance between pro-fibrotic miRNAs and anti-fibrotic miRNAs seems to be the key to repairing the injured myocardium. In this review, we highlight the most relevant miRNAs involved in post-MI LVR.

## 2. miRNAs and Post-MI Remodeling

### 2.1. LVR Mechanisms

Cardiac remodeling involves three intricate and complex processes (apoptosis, hypertrophy, and fibrosis), of which fibrosis is the main component of post-MI LVR. Cardiac fibrosis evolves as a pathological development of healing if the injury is severe, repeated or the healing response itself becomes dysregulated. Fibrosis represents an uncontrolled accumulation of extracellular matrix (ECM) components (collagen type I and III, elastin, and glycosaminoglycans) produced by fibroblasts and myofibroblasts [8]. In LVR, cardiomyocytes are replaced by connective tissue, leading to the formation of a permanent scar and subsequent loss of function of the affected area.

LVR is initiated in the acute phase of MI when macrophages release factors that stimulate fibroblasts. Most frequently, this factor is transforming growth factor-beta (TGF-β), secreted as a latent complex, unable to associate with its receptors. TGF-β activation involves cleavage of the C-terminal and N-terminal pro-domains by extracellular proteases, such as furin, plasmin, or Matrix Metalloproteinases (MMP)-2 and MMP-9 [9]. Other cellular pro-fibrotic mediators include connective tissue growth factor (CTGF), platelet-derived growth factor (PDGF), tumor necrosis factor alfa (TNF-α), interleukin 10 (IL-10,) and Angiotensin II (*AngII*) [10].

TGF-β signaling is the fundamental pathway in heart fibrosis. TGF-β (as well as CTGF) binds to type I and type II TGF-β receptors to form a complex, which initiates a signaling cascade that leads to phosphorylation of Smad (mothers against decapentaplegic homolog) proteins [11]. The activation of TGF-β receptor I (TGFβRI) phosphorylates Smad 2 and 3 transcription factors in the cytoplasm [11]. The activated Smads form complexes with Smad 4 and translocate to the nucleus, where they induce myofibroblast formation and activate pro-fibrotic genes for collagen (collagen type 1, alpha-1 gene, collagen type 1, alpha-2 gene, collagen type 3, alpha-1 gene, etc.), elastin (*ELN*), fibrillin (*FBN*), and alfa smooth muscle actin (α-*SMA*) synthesis [11]. Of note, Smad 7 inhibits TGF–β signaling pathway by interfering with the activation of the Smad 2/3 complex [11].

Besides the Smad canonical pathway, TGF-β activates many non-canonical signaling pathways, such as ERK (extracellular signal-regulated kinase), MAPK (mitogen-activated protein kinase), AKT (protein kinase B), and NF-κB (nuclear factor kappa-light-chain-enhancer of activated B cells) pathways. Ultimately, this leads to proliferation, differentiation, and the activation of fibroblasts, which deposit ECM components in the surrounding tissue [11,12].

### 2.2. miRNAs in LVR after MI

A large number of miRNAs have been associated with the process of fibrosis. In Table 1, we highlight the most significant miRNAs involved in post-MI LVR (Table 1).

MicroRNA modulation of collagen production through canonical/Smad and non-canonical TGF-β pathway is instrumental for understanding post-MI fibrosis. Some miRNAs (miR-7, miR-29, miR-34a, miR-92, miR-125b, miR-133a, miR-208, miR-210, miR-370, and miR-1254) target only the canonical/Smad pathway, while others (miR-19a/b, miR-150, miR-199b, miR-223, let-7) target non-Smad pathways at multiple levels (Figure 1 and Figure 2).

MiRNAs colored in green have an anti-fibrotic function; miRNAs colored in red have a pro-fibrotic role. Dash-line arrows indicate the translocation of proteins from the cytoplasm to the nucleus or vice versa; solid line arrows indicate signaling cascades; yellow stars indicate transcription factors. α-SMA, alfa smooth muscle actin, AZIN1, antizyme inhibitor 1; c-Myb, member of proto-oncogene family Myb; *COL1A1*, collagen type 1, alpha 1 gene; *COL1A2*, collagen type 1, alpha 2 gene; CTGF, connective tissue growth factor; *ELN*, elastin gene; ENG, endoglin; *FBN*, fibrillin gene; RI, transforming growth factor-beta receptor I; RII, transforming growth factor-beta receptor II; Smad 2, 3, 4, 7, 2/3, SMAD family members; Snai1, snail family zinc finger 1; Sp1, specificity protein 1; TGF-β, transforming growth factor-beta.

MiRNAs can either directly or indirectly modulate collagen synthesis and other molecules involved in the progression of cardiac fibrosis. Most of the miRNAs involved in post-MI LVR modulate the TGFβ signaling pathway through fibroblast activation, while others (e.g., miR-7, miR-146, miR-199, miR-214) are linked to fibrosis through cardiomyocytes’ apoptosis. Interestingly, the activity of these miRNAs is cell-specific: miR-15, miR-133a, miR-146, miR-210, miR-370 in cardiomyocytes and fibroblasts; miR-21, miR-26a in endothelial cells, miR-155 in macrophages; and, exceptionally, miR-19a/19b in cardiomyocytes, fibroblasts, and macrophages.

However, although TGFβ is the central signaling in post-MI LVR fibrosis, it appears that only modulating the expression of miRNAs associated with this pathway is not enough to achieve a proper healing response. Furthermore, none of these miRNAs can be considered by itself a bona fide prognostic biomarker or a therapeutic target.

### 2.3. Multifunctional miRNAs

Individual miRNAs can modulate signaling pathways in more than one cell type, suggesting they regulate multiple target genes and have various functions in post-MI LVR [66]. Most of these miRNAs have a role in cardiomyocyte death or hypertrophy, while some control the angiogenesis process and/or the inflammatory response (Table 2).

The advent of microRNA research in MI prompted novel therapeutic approaches aiming to modulate cardiac remodeling through in vivo manipulation of miRNAs levels microRNA-mimics and antagomirs. miRNA-based therapeutics have the potential of becoming a whole new class of drugs in the treatment of MI. The ideal cocktail of miRNAs to prevent post-MI LVR would consist of specific multifunctional miRNAs targeting multiple pathways of this process. For example, combining miR-mimics for miR-19a/19b, miR-26, miR-133a, and miR-210 with antagomirs for miR-21, miR-92, and miR-155 might lead to a better outcome of MI events, with less scarring tissue, less inflammation, and increased cardiomyocyte regeneration leading to improved cardiac function.

### 2.4. miRNAs and Remodeling Induced by Pressure Overload

Cardiac remodeling is mediated by miRNAs that exert different functions by regulating various downstream pathways, depending on whether the trigger is MI or pressure overload (Table 3).

Several miRNAs (miR-21, miR-34, miR-199b) have similar roles in cardiac fibrosis post-MI or induced by TAC/*AngII* and even share some of the downstream targets: (miR-34 and miR-21) in the two fibrosis settings. At the same time, miR-15b and miR-26a have opposing roles in the two LVR settings. Post-MI, miR-15b induced involves modulation of apoptosis and hypertrophy, while post-fibrotic overload fibrosis utilizes the TGFβ pathway. MiR-26a exploits TGFβ canonical and non-canonical (through NF-κB) pathways to reach divergent effects on fibrosis in the two scenarios. Interestingly, while miR-19a/b has a pronounced anti-fibrotic effect post-MI, no changes in its plasma levels were detected in a TAC mouse model. The two members of the Let-7 family also have opposing effects on cardiac remodeling and target different genes. Let-7c has a pro-fibrotic effect by stimulating proliferation and differentiation in fibroblasts and apoptosis in cardiomyocytes, while let-7i has an anti-fibrotic effect modulation of inflammation and collagen genes’ expression.

Multiple questions arise related to these paradoxical findings. Could they be justified solely by the mechanism which initiated the remodeling? Are these miRNAs secreted in antagonistic pairs through common regulatory mechanisms? Do miRNA targeting specificity changes in the different cellular contexts and triggering pathophysiological MI mechanisms? Although the molecular and cellular processes underlying miRNA-related cardiac fibrosis in both situations are rapidly disentangled, much remains to be learned regarding the role of these molecules.

## 3. miRNAs as Prognostic Biomarkers of Post-MI LVR

Besides their diagnostic use, MI-associated miRNAs are also considered as potential prognostic biomarkers. Next, we examine the emerging prognostic role of several circulating miRNAs in post-MI LVR (Table 4).

miR-1 was recently proposed as a prognostic biomarker for post-MI LVR [87]. Ma and colleagues investigated 80 patients with first-time MI with ST-elevation. One and six months after the MI event, the patients underwent cardiac magnetic resonance (CMR) with measurement of LVR parameters and the changes in left ventricle ejection fraction (%LVEF), left ventricle end-diastolic volume (%LVEDV), and left ventricle end systolic volume (%LVESV). With LVR defined as a minimum of 10% increase in LVEDV, miR-1 at admission showed a positive correlation with %ΔLVEDV and %ΔLVESV, but not with %ΔLVEF in 22 patients. In addition, miR-1 was correlated with clinical biomarkers (especially N-terminal pro-brain natriuretic peptide—NT-proBNP, but also with creatine kinase MB—CK-MB and cardiac troponin—cTn also). Therefore, an increase in plasma miR-1 levels was associated with a negative outcome regarding LVR after MI.

This study contradicts the findings of Grabmeier et al. In 2017, the study group investigated the prognostic value of four miRNAs (miR-1, miR-21, miR-29b, and miR-92) in 44 patients with MI from the SITAGRAMI study [81]. Blood samples were collected on days 4, 9, and 6 months after MI, CMR assessment (infarct volume (IV), LVEF, and LVEDV) was obtained on day 4 (± 2) and six months after MI, and absolute changes in these parameters were calculated. The serial measurements for these miRNAs revealed the same pattern for miR-1 and miR-21 (increase on day 4 and 9 with a decrease to day 4 values at the 6-month follow-up), an ascending pattern for miR-29b, and no change for miR-92 levels. Of the four miRNAs, miR-1 and miR-29b correlated with lower infarct volume and only miR-29b with lower LVEDV. Thus, an increase in miR-1 and miR-29b concentrations was associated with a favorable outcome concerning LVR after MI.

In 2011, Zile et al. described the first temporal profile of 5 miRNAs (miR-1, miR-21, miR-29a, miR-133a, and miR-208) in 12 patients with MI [89]. Blood samples were collected on days 2, 5, 28, and 90 post-MI. LVEDV was measured by echocardiography on days 1, 5, 28, and 90 post-MI and showed a time-dependent increase. Compared with controls, the miR-1 level decreased on day 2, increased on day 5, and remained unchanged through day 90 post-MI. MiR-21 decreased on day 2, increased on day 5, and returned to normal on day 90 post-MI. MiR-29a levels were not modified on day 2, increased on day 5 but returned to normal levels by day 90. MiR-133 and miR-208 were unchanged on day 2, increased on day 5, and remained elevated through day 90. Of note, only day 5 miR-29a levels showed a positive association with day 90 LVEDV. These findings suggest that miR-1 has no prognostic value for post-MI LVR and that early miR-29a increase in the blood correlates with a negative outcome of post-MI LVR.

The promising role of miR-21 as a biomarker of LVR has also been investigated in human trials. The study group of Dubois explored the role of superoxide dismutase 2 (SOD2), a major antioxidant enzyme, and three circulating miRNAs (miR-21, miR-222, and miR-23) in patients with MI [91]. They used blood samples from the Remodelage Ventriculaire–2 (REVE-2) multicenter study patients, which enrolled 246 probands with a first anterior wall Q-wave MI. The patients underwent periodical (three months and one year) echocardiographic evaluation and blood sampling and were divided into a *no LVR* or a *high LVR* group, according to the echocardiography results. The circulating levels of the three miRNAs and their target, SOD2, were associated with high LVR after myocardial infarction.

Gao and colleagues found a positive association between miR-21 serum levels and the neutrophil-lymphocyte ratio in 184 patients with MI [92]. The 30-day survival rate of patients with high miR-21 serum levels and the elevated neutrophil-lymphocyte ratio was significantly lower than patients with low miR-21 expression and low neutrophil-lymphocyte ratio, suggesting that miR-21 can be used as an independent predictor for survival post-MI.

Liu et al. analyzed 198 MI patients to assess the possible use of miR-21 or miR-146a as LVR predictors [93]. All patients were evaluated by echocardiography (LVEF, LVEDV, LVESV) on day five and one year after percutaneous coronary intervention (PCI), and blood samples were collected on day five after PCI. LVR was defined as an increase of at least 20% in LVEDV one year after MI (*n* = 56). They also evaluated plasma levels of CK-MB, cTnI, C Reactive Protein, NT-proBNP, leukocytes and calculated the estimated Glomerular Filtration Rate. MiR-21 correlated with cTnI, and miR-146 was associated with CRP and leukocyte count. The results showed that both miR-21 or miR-146a could be used as early biomarkers of LVR, and their combination might have a higher predictive power. On day five, both miRNAs showed a higher concentration in the LVR group compared to controls. While for miR-21, which has a proven pro-fibrotic role, these findings are expected, for miR-146, which has an anti-fibrotic role, these results are rather surprising and might be explained by its proinflammatory function.

Maciejak et al. used digital PCR to monitor miR-30a-5p levels in plasma and serum. They found significant elevations on admission in MI patients who had developed heart failure at the 6-month follow-up (defined by LVEF ≤ 50%, NT-proBNP ≥ 150 pg/mL). In this study, the role of miR-30 was described as pro-fibrotic. A GO (Gene Ontology) and KEGG (Kyoto Encyclopedia of Genes and Genomes) analysis identified fibroblast growth factor signaling via the PI3K/AKT pathway, which plays a crucial role in LVR as a key pathway miR-30a-5p [94]. However, these results should be interpreted cautiously, given the small sample size (*n* = 14) and the cut-off values for defining the HF and non-HF groups.

In 2014, Pin et al. assessed the prognostic significance of miR-34a and miR-208b levels in blood sampled in admission on a large cohort (*n* = 359) of MI patients [95]. Echocardiography was performed after admission (day 1–4) and at the 6-month follow-up. LVR parameters were measured, and an increase of more than 10% in LVEDV defined the cardiac remodeling group (*n* = 116). Both miRNAs showed higher blood levels in the LVR group compared with the non-LVR group. The primary endpoint consisting of cardiovascular death or heart failure (clinical symptoms, LVEF < 40% and high NT-proBNP levels) was noted in 83 patients and associated with significantly increased miR-208b and miR-34a levels. Furthermore, the combination of these two plasma miRNAs outperformed NT-proBNP in both LVR and endpoint prediction after MI.

Zhang et al. also proposed a prognostic role of miR-92a for LVR after MI [41]. The study cohort consisted of 60 MI patients who underwent PCI. Echocardiography was performed before and at three months after PCI and included EF, LVEDV, LVESV, and LV wall thickness measurements. High blood levels of miR-92 on admission positively correlates with LVEDV, LVESV, and LV wall thickness and negatively with LVEF at three months. This confirms the pro-fibrotic role of miR-92 in the previous studies but contradicts the findings of Grabmeier et al., which found no detectable changes in miR-92 levels after MI [88].

While previous studies suggested that miR-133 and miR-423-5p can be used as heart failure biomarkers, Bauters et al. showed that these miRNAs were not associated with LVR echocardiographic parameters (LVEF, LVEDV, and LVESV), brain natriuretic peptide, cTnI, or C reactive protein over one year [99]. The study population consisted of 246 patients with anterior Q-wave MI, so an inadequate sample size could not explain the result, especially since other miRNAs (e.g., miR-21) investigated in blood samples from the same study (REVE-2) showed various degrees of association. The result was confirmed by Zile et al. [96].

The first study to report the use of miR-150 as a prognostic biomarker for LVR emerged in 2013 [97,101]. The authors used an in-silico approach to select which miRNAs would most likely be involved in the development of LVR and retrieved 13 genes known to be associated with LVR and, later on, 265 miRNAs targeting these genes. Then, using an interaction network, they selected ten miRNAs, out of which, based on blood analysis findings, two miRNAs emerged as possible candidates: miR-150 and miR-101. Finally, they chose miR-150 due to its deregulation in cardiac hypertrophy. Subsequently, the study group demonstrated that miR-150 is overexpressed in the infarcted tissues of mice subjected to experimental (coronary ligation) MI.

Furthermore, they showed that miR-150 regulates LVR by inhibiting C-reactive protein and adrenergic receptor beta 1 expression. Of note, the authors used a derivation cohort (60 patients divided into two groups based on whether they presented with LVR, as assessed by echocardiography—Δ End Diastolic Volume), as well as a validation cohort (the subjects underwent cardiac magnetic resonance to determine whether LVR is present after MI). They reported a higher predictive power of miR-150 for LVR than NT-proBNP, or a multiparameter model including serum markers at admission, age, and sex. They concluded that LVR is associated with low circulating levels of miR-150 after the first MI [97].

Later that year, the same authors published data evaluating a signature combination of four miRNAs (miR-16/27a/101/150) to predict post-MI LVR [90]. One hundred fifty patients with MI were enrolled in the study, blood samples were obtained at discharge (to determine NT-proBNP plasma levels and the four miRNAs), but this time the wall motion index score as an indicator for LV systolic dysfunction was applied. They built two multi-variable models to assess LV contractility: one model included clinical variables, including NT-proBNP, and the second model included all the parameters of model 1 and the four miRNAs. Patients with anterior MI or a history of MI and elevated NT-proBNP were at high risk of impaired LV contractility. Patients with low levels of miR-150/101 or elevated levels of miR-16/27a were also at increased risk of poor LV contractility. Statistical analysis confirmed that the panel of four miRNAs improved the prognostic value of the multi-variable clinical model, including NT-proBNP.

The two papers published by Devaux and colleagues correlate with data supporting the role of miR-150 in cell proliferation, migration, and differentiation and designate miR-150 as a promising prognostic biomarker for the clinical outcome post-MI; nevertheless, more complex studies are required to refine this hypothesis further.

The role of miR-208 as a MI prognostic value was investigated in four studies. As mentioned above, Lv et al. confirmed the prognostic power of miR-208b and miR-34a in a large sample of patients [95].

In 2017, Liu and colleagues proved that miR-208 could serve as a predictor for post-MI LVR [98]. The study cohort consisted of 100 MI patients (of which 59 underwent PCI), 80 patients with unstable angina, and 80 healthy controls. Venous blood samples were collected on admission and after PCI. Echocardiographic assessment of cardiac function (LVEDV, LVESV, and LVEF) was completed on admission and at the 6-month follow-up. The LVR group was defined as at least a 10% increase in LVEDV from baseline (*n* = 14). Major clinical cardiovascular events (MACE) were evaluated in all MI patients who underwent PCI (*n* = 20). Circulating miR-208b levels were significantly higher in MI patients than in unstable angina patients and healthy controls. Additionally, miR-208b level was increased in three-vessel coronary artery disease (CAD) patients compared to single- and two-vessel CAD. Last but not least, miR-208b blood levels were higher in the LVR and MACE groups. Interestingly, the plasma level of miR-208b was significantly decreased after PCI compared to levels before PCI.

In 2018, Alavi-Maghaddam et al. published the results of a pilot study in which they investigated whether miR-208b can be used as a predictor of 6-month survival time after MI [99]. The study enrolled 21 MI patients with ST-elevation, among which seven (the non-survivor group) died within six months after MI. The plasma levels of circulating miR-208b were significantly higher compared to the survivor group. This preliminary, small study results showed that miR-208b could serve as an indicator of survival six months after MI.

Collectively, these findings support the notion that miR-208b has a negative correlation with the improvement of cardiac function and may be a potential predictor of post-MI LVR.

In a study conducted by de Gonzalo-Calvo and colleagues, miR-1254 showed a significant association with cardiac magnetic resonance (CMR) variables (positive with LV ejection fraction and negative with LV end-diastolic volume index and LV end-systolic volume index) [100]. They identified miR-1254 as an independent predictor of LVR in patients after one week, respectively, and six months (with multivariate adjustment including clinical, CMR variables, hs-troponin-T, and NT-proBNP) after MI with ST-elevation.

The prognostic value of miR-1 has been investigated in three studies, all with a different outcome. Given its anti-fibrotic and antihypertrophic role, high levels of miR-1 should be associated with a lack of cardiac remodeling [102]. The same dilemma applies to miR-29 and miR-30a studies. For miR-29, two studies show opposing results, although both suggest miR-29 can be used as a predictor for MI. While overexpression of miR-30a in rats had a cardioprotective effect, in human trials, it was associated with LVR.

The contradictory conclusions of these studies might be explained mainly by the different methods used to assess (CMR vs. echo) and define LVR (absolute change vs. percentage, LVDEV vs. LVEF, echo parameters vs. paraclinical markers), although the time frames for clinical and paraclinical evaluation (three months vs. six months vs. one year) and the diverse inclusion or exclusion criteria (STEMI vs. all types of MI) should also be taken into consideration.

To solve some of these impediments, we advance some suggestions from a clinical perspective. First, we would like to emphasize the necessity for a standardized, easy-to-use definition for left ventricle remodeling. The use of echocardiography should become a standard since it is accessible in most hospitals, it is non-invasive, easily repeatable, safe, quick, and does not require ultra-specialized training. Combining three, easy to measure parameters, such as LVED, LVESV, and LVEF provides a clear image of the cardiac function post-MI. Lately, the global longitudinal strain has emerged as a less operator-dependent parameter for LVR, but further evidence is required [103]. In addition, a standardized timeline (on admission, at discharge, at the 1-month, 6-month, and 1-year follow-ups, and after that, annually) for these evaluations should be implemented.

As for miRNA evaluation, we propose an early (either on admission or at discharge) single-point measurement of miRNAs with putative prognostic power. This would overcome the typical challenges associated with miRNA studies, such as the time-dependent changes of the circulant plasma following MI. Furthermore, one should remember that almost 50% of patients are non-compliant because of forgetfulness, medication costs, inability to get a prescription/consultation, poor doctor-patient relationship, or adverse side-effect of medication [104,105]. Therefore, the sooner we can appreciate the development post-MI, the better we can control it.

For miR-21, miR-150, and miR-208, the studies arrive at the same conclusion, underlining the need for further, more complex analyses aiming to refine their prognostic and therapeutic value. Further confirmations on much larger cohorts are required for the rest of the miRNAs reviewed, including miR-34 and miR-146. Lastly, we want to underline that combining multiple miRNAs (for example, miR-16/27a/101/150, miR-21, and miR-146 or miR-34 and miR-208) has higher predictive power for LVR compared to a single miRNA measurement.

## 4. miRNA as Therapeutic Targets

Although the last decade’s research has vastly expanded our knowledge on the therapeutic applications of microRNAs in acute myocardial infarction, the clinical translation still faces many challenges; one of the most important is the targeted delivery of miRNAs. Delivering microRNAs into an infarcted heart is not a simple task since multiple issues have to be taken into consideration for a successful approach: the timing of administration, the route of administration (intracardiac injection in the infarcted area/periphery of infarcted area, intravenous administration), the packaging of the small ARN (exosomes, lentiviral, synthetic nanoparticles, nanocarriers).

Li et al. used lentiviral vectors to overexpress miR-7a/b to reduce fibrosis apoptosis and cardiac remodeling and improve cardiac function in a mouse model of myocardial infarction [13].

Direct intracardiac injection of agomiR-21 and agomiR-146a conjugated with cholesterol moieties has been shown to significantly reduce apoptosis in the infarcted hearts of Kunming mice [26].

Chen et al. injected intramyocardially miR-133-conditioned mesenchymal stem cells and reported a reduction of apoptosis and the infarct size and a significant increase in cardiac function in a rat model of acute myocardial infarction an effect presumably mediated by the local release of miR-133 [106].

The administration of antimiR-21 by intracoronary perfusion in pigs subjected to transient percutaneous occlusion of the left coronary artery suppressed pathological post-MI fibrosis and the inflammatory response in macrophages [107]. Interestingly, the use of anti-miR-21-coated stents leads to an efficient reduction of in-stent restenosis, with no significant off-target side effects [108,109]. Bellera and colleagues showed that a single, early (at the reperfusion time) intracoronary administration of encapsulated in polylactidecoglycolide anti-miR-92 stimulates angiogenesis and prevents LVR in an adult pig model of AMI, without evidence of systemic or local adverse effects [42].

One of the most frequently used delivery methods is represented by exosomes. Exosomes are 30–100 nm wide, CD9/63/81 positive extracellular vesicles can transfer proteins, mRNAs, and microRNAs between cells, thus being involved in intercellular communication [110]. Due to their low immunogenicity profile, exosomes are well tolerated, and, depending on their surface, antigens can target specific cells, properties that make them ideal therapeutic delivery tools.

Song et al. injected miRNA-21-enriched exosomes (isolated from conditioned HEK293T cells) directly into the infarcted area and documented a rapid (four hours) uptake of miR-21 in both cardiomyocytes and endotheliocytes, followed by a significant attenuation of apoptosis and improvement of cardiac function [111]. Ma et al. used mesenchymal cells-derived exosomes injected intramyocardially to deliver miR-132 in infarcted mouse hearts and described an improved vascularization in the peri-infarcted area, with significant enhancement of cardiac function [112]. Early intravenous administration of exosomes derived from miR-146a-conditioned adipose-derived stem cells significantly reduced the cardiac damage and improved cardiac function in a rat model of acute myocardial infarction [113]. Zhao et al. injected intramyocardially exosomes derived from mesenchymal stromal cells and showed that exosomal miR-182 reduced cardiac inflammation and infarct size in a mouse model of myocardial infarction [114]. Charles et al. reported a significant reduction in the infarcted size and improved cardiac function after IV administration of MSC-derived exosomes as early as day seven post-infarction [115].

However, the therapeutic use of exosomes is still hampered by its low and variable recovery yield from cell cultures, their widely heterogeneous size, and the cumbersome purification procedures [116,117]. Alternatives to the exosome, such as targeted modified nanoparticles and low molecular weight heparin (LMWH) nanocarriers, have been recently proposed. Hyaluronan-sulfate miR-21-carrying anionic targeted at macrophages in the infarcted area turned them in reparative mode and significantly reduced fibrosis and apoptosis, with enhanced cardiac function [118]. Even more exciting, LMWH nanocarriers loaded with anti-miR-1 overcame micro-thrombotic obstruction and reduced cardiomyocyte apoptosis, cardiac fibrosis, and promoted tissular repair [119].

## 5. Conclusions

As outlined in this article, miRNAs are essential regulators of post-MI cardiac remodeling. MiRNAs can control the post-MI cardiac fibrotic process in its entirety, thus providing new therapeutic targets for fibrosis therapy. The manipulation of multifunctional miRNAs into therapeutic drugs may hold the key to controlling a complex disease such as cardiac remodeling. Furthermore, they could serve as novel prognostic biomarkers of post-MI LVR, which can be used in the long-term follow-up of MI patients. With this review, we hope to shine some light on the miRNA-based diagnostic, prognostic, and therapeutics of cardiac remodeling after myocardial infarction.

## 6. Clinical Perspective

The management of MI patients needs novel, more sensitive, and specific diagnostic, prognostic, and predictive biomarkers; some of the microRNAs discussed in the present review fulfills these criteria. Furthermore, given their function in the infarcted tissue, they might also qualify as therapeutic tools.

For several miRNAs, the diagnostic role in MI has been well established, even with the same or higher sensitivity and specificity than traditional biomarkers.

A prognostic biomarker can help the physician predict the outcome and overall survival of a MI patient receiving standard or no treatment. A predictive biomarker aids the clinician in making tailored treatment decisions (drug or dosage selection). So far, miRNAs have been intensively investigated as predictive biomarkers, mostly in different types of cancer. Identifying miRNAs with predictive power post-MI would be a valuable way to distinguish which patients are likely or unlikely to benefit from a particular treatment. For example, a patient with high plasma levels of pro-fibrotic miRNAs, such as miR-21, miR-155, and miR-208, might benefit from an increased number of follow-up visits, regular cardiac function evaluation (both non-invasive and invasive, if necessary), higher dosage for drugs that prevent LVR (angiotensin-converting-enzyme inhibitors and angiotensin II receptor blockers), more extended periods of administration of a high dosage of statins or dual antiplatelet therapy in patients, cardiac rehabilitation and home monitoring.

The therapeutic role of miRNAs with post-MI LVR is the most fascinating. Therapy with anti- or ago-miRs has been evaluated in mice, rats, or pig models, but, so far, not in humans. Further studies are needed to establish standardized miRNA (cocktail) therapeutic protocols, addressing obstacles such as design optimization, method and timing of delivery, and, most important, defining targeted delivery to specific cardiac cells without local or systemic side effects.

However, we are halfway down the road by identifying diagnostic and prognostic miRNAs and testing some therapeutic ones in vivo.

## Figures and Tables

**Figure 1 diagnostics-11-01675-f001:**
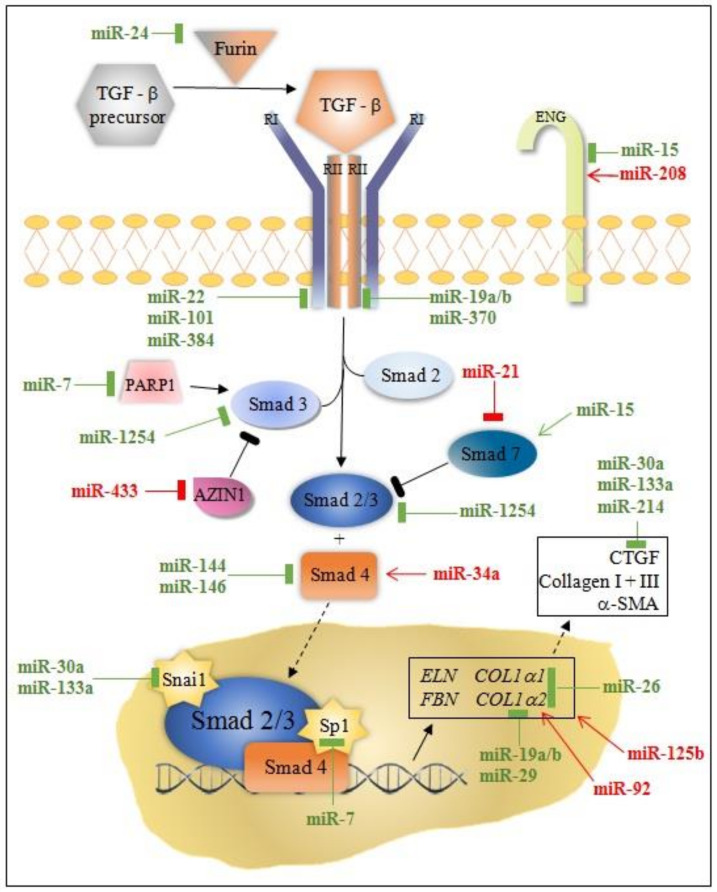
Schematic representation of the canonical TGF-β signaling pathway and related miRNAs.

**Figure 2 diagnostics-11-01675-f002:**
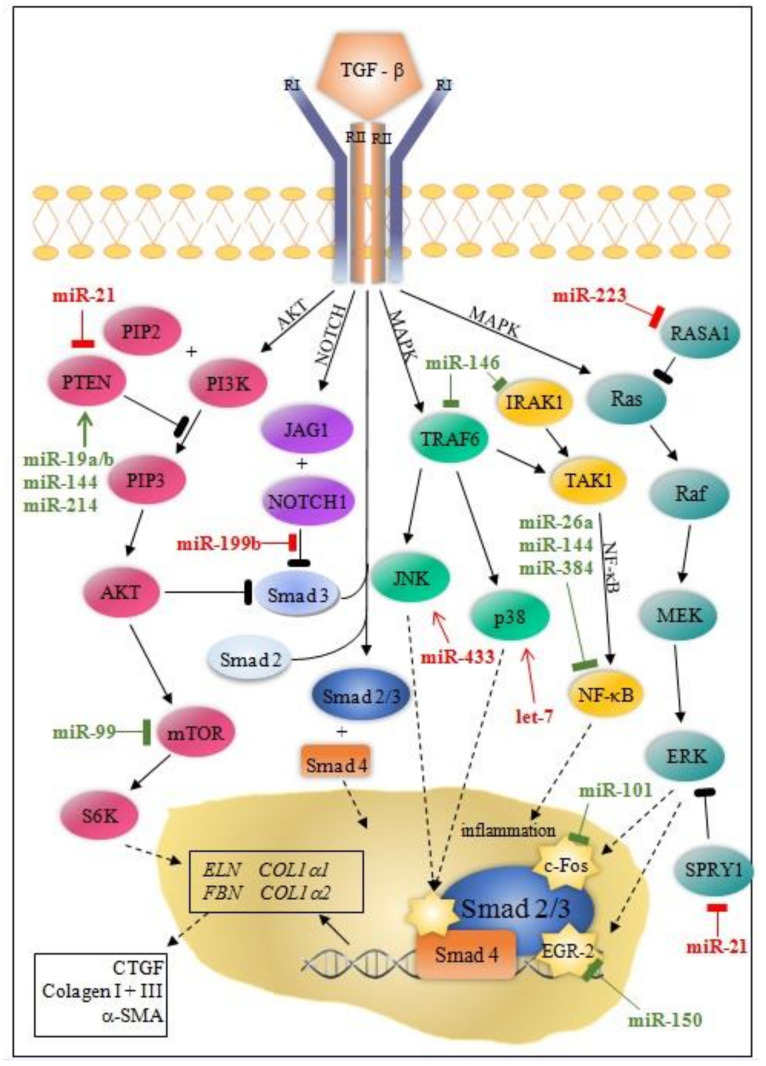
Schematic representation of the non-canonical TGF-β signaling pathway and related miRNAs. MiRNAs colored in green have an anti-fibrotic function; miRNAs colored in red have a pro-fibrotic role. Dash-line arrows indicate the translocation of proteins from the cytoplasm to the nucleus or vice versa; solid line arrows indicate signaling cascades; yellow stars indicate transcription factors. α-SMA, alfa smooth muscle actin, AKT, protein kinase B; c-fos, Fos Proto-Oncogene, AP-1 transcription factor subunit, *COL1A1*, collagen type 1, alpha 1 gene; *COL1A2*, collagen type 1, alpha 2 gene; CTGF, connective tissue growth factor; EGR-1, early growth response protein 1; *ELN*, elastin gene; ERK, extracellular signal-regulated kinase, *FBN*, fibrillin gene; IRAK1, interleukin-1 receptor-associated kinase 1; JAG1, jagged 1 protein; JNK, c-Jun N-terminal kinases; MAPK, mitogen-activated protein kinase; MEK, mitogen-activated protein kinase kinase; mTOR, mammalian target of rapamycin protein; NF-κB, nuclear factor kappa-light-chain-enhancer of activated B cells; NOTCH1, Notch receptor 1; P38, p38 mitogen-activated protein kinases; PI3K, phosphoinositide 3-kinases; PIP2, phosphatidylinositol 4,5-bisphosphate; PIP3, phosphatidylinositol (3,4,5)-trisphosphate; PTEN, phosphatase and tensin homolog protein; Raf, Proto-Oncogene, Serine/Threonine Kinase; Ras, Ras Proto-Oncogene, GTPase family; RI, transforming growth factor beta receptor I; RII, transforming growth factor beta receptor II; S6K, ribosomal protein S6 kinase beta-1; Smad 2, 3, 4, 2/3, SMAD family members; TAK1, mitogen-activated protein kinase kinase kinase 7 or transforming growth factor-beta-activated kinase 1; TGF-β, transforming growth factor beta, TRAF6, tumor necrosis factor receptor associated factor 6.

**Table 1 diagnostics-11-01675-t001:** Summary of miRNAs involved in cardiac fibrosis after MI.

miRNA	Effect on Fibrosis	Target	Effector Cells	Platforms	Reference
miR-7a/b	Anti-fibrotic	*Sp1* *PARP-1* *TGFβ*	CM	MI mouse H9C2 cell line cardiomyoblast	[13]
miR-9	Pro-fibrotic	*Follistatin-like 1*	CM	MI mouse	[14]
miR-15 family	Pro-fibrotic	*ARL2* *BCL2*	CM	MI pig MI mouse	[15]
miR-19a/19b	Anti-fibrotic	*COL1A1, COL3A1, ELN, FBN1, TGFβRII, PTEN*	CM F M	MI mouse Human heart tissue	[16]
miR-21	Pro-fibrotic	*Smad7* *Spry1* *ERK*	F CM M	MI mouse and pig NRCFs	[17,18,19,20,21,22,23,24,25,26,27]
miR-22	Anti-fibrotic	*TGFβRI*	F	MI mouse NRCFs	[28]
miR-24	Anti-fibrotic	*Furin*	F	MI mouse NRCFs	[29]
miR-26a	Pro-fibrotic	*Smad1, BMP* *NF-κB* *COL1A1* *COL1A2*	EC	MI mouse HUVECs Human trials	[30]
miR-29	Anti-fibrotic	*FBN1, COL1A1* *COL1A2, COL3A1*	F	MI mouse RCFs	[31,32,33]
miR-30a-5p	Anti-fibrotic	*CTGF* *TGFβ*	F CM EC	MI Rat NRVMs NRCFs	[34,35,36,37,38,39]
miR-34a	Pro-fibrotic	*Smad4*	F	MI mouse	[40]
miR-92	Pro-fibrotic	*Collagen 1* *IL-6, TNF-* *α* *BCL-2*	F CM M	MI rat Human trials	[41,42]
miR-99	Anti-fibrotic	*mTOR/S6K*	CM	MI mouse NMVMs	[43]
miR-101	Anti-fibrotic	*TGFβR1* *c-Fos*	F	MI rat NRCFs	[31,44,45]
miR-125b	Pro-fibrotic	*TGFβ* *Apelin* *p53* *SFRP5*	F	MI mouse HCFs Human heart tissue	[46,47]
miR-133a	Anti-fibrotic	*CTGF* *Snai1* *SRF*	CM F	MI mouse NRCFs	[48,49]
miR-144	Anti-fibrotic	ZEB1/LOX axis	F	MI mouse NRCFs	[50,51]
miR-146	Anti-fibrotic	*Smad4* *IRAK1* *TRAF6*	F CM	MI mouse NRCMs CDC	[52]
miR-150	Anti-fibrotic	*EGR2* *P2 × 7R* *CXCR4*	F CM M	MI mouse Human trials	[53,54]
miR-155	Pro-fibrotic	*SOCS1*	M	MI mouse PCMs	[55]
miR-199b	Pro-fibrotic	*NOTCH1* *JAG1* *DYRK1A*	CM	MI mouse	[56]
miR-208a	Pro-fibrotic	*Endoglin*	F	MI rat RCFs	[57]
miR-210	Anti-fibrotic	*EFNA3* *PTP1* *DAPK1* *CTGF*	CM F	MI mouse Mouse HL-1 CMs	[58]
miR-214	Anti-fibrotic	*PTEN*	CM	MI rat	[59]
miR-223	Pro-fibrotic	*RASA1*	F	MI mouse Cultured cardiac fibroblasts	[60]
miR-370	Anti-fibrotic	*TGFβRII*	F CM	MI rat RCFs	[61]
miR-384	Anti-fibrotic	TGFβRI/Wnt(3a)/NF-κb	F	I/R rats NRCFs	[62]
miR-433	Pro-fibrotic	*AZIN1* *JNK1*	F	MI mouse NRCFs	[63]
Let-7c	Pro-fibrotic	*OCT4* *SOX2* TGFβRIII/p38	F CM	MI mouse AMCFs MI pig	[64,65]

AMCFs, adult mouse cardiac fibroblasts; *ARL2*, ADP-Ribosylation Factor-Like Protein 2; *BCL2*, Apoptosis Regulator BCL2; *BMP*, bone morphogenetic protein; *c-fos*, Fos Proto-Oncogene; CDC, cardiosphere-derived cells; CM, cardiomyocyte; *COL1A1*, collagen type 1, alpha 1 gene; *COL3A1*, collagen type 3, alpha 1 gene; *CTGF*, connective tissue growth factor; *DAPK1*, Death-associated protein kinase 1; *DYRK1A*, dual specificity tyrosine-phosphorylation-regulated kinase 1A; EC, endothelial cell; *EFNA3*, Ephrin A3; *EGR-2*, early growth response protein 2; ELN, elastin gene; *ERK*, extracellular signal-regulated kinase; F, fibroblast; *FBN*, fibrillin gene; HCFs, human cardiac fibroblasts; HUVECs, Human Umbilical Vein Endothelial Cells; I/R, ischemia/reperfusion; *IL-6*, Interleukin 6; *IRAK1*, interleukin-1 receptor-associated kinase 1; *JAG1*, jagged 1 protein; *PARP1*, Poly [ADP-ribose] polymerase 1; M, macrophage; MI, myocardial infarction; *mTOR*, mammalian target of rapamycin protein; NMVMs, neonatal mice ventricular myocytes; *NOTCH1*, Notch receptor 1; NRCFs, Neonatal rat cardiac fibroblasts; NRCMs, neonatal rat cardiac myocytes; *OCT4*, octamer-binding transcription factor 4; *P2RX7*, P2X purinoceptor 7; PCMs, primary cultured macrophages; *PTEN*, phosphatase and tensin homolog protein; PTP1, Protein-tyrosine phosphatase 1; RASA1, Ras GTPase activating protein 1; RCFs, rat cardiac fibroblasts; *S6K*, ribosomal protein S6 kinase beta-1; *SFRP5*, secreted frizzled-related protein 5; *Smad 1, 4, 7,* SMAD family members; *Snai1*, snail family zinc finger 1; *SOCS1*, Suppressor of cytokine signaling 1; *Sp1*, specificity protein 1; *Spry1*, sprouty homologue 1; *SRF*, Serum Response Factor; *TGFβ*, transforming growth factor beta; *TGFβRI*, transforming growth factor beta receptor I; *TGFβRII*, transforming growth factor beta receptor II; *TNF-α*, tumor necrosis factor alfa; *TRAF6*, tumor necrosis factor receptor associated factor 6.

**Table 2 diagnostics-11-01675-t002:** Summary of multifunctional miRNAs in myocardial infarction.

miRNA	Fibrosis	Hypertrophy	Apoptosis	Angiogenesis	Inflammation	Reference
miR-19a/19b	−	− (Ang II model)	−	−	−	[16,67,68]
miR-21	+	+	−	+	−	[19,20,21,22,23,24,25,26,27,69]
miR-26	−	− (TAC, *AngII* model, exercise-induced)	−	−	−	[30,70,71,72,73]
miR-92	+	+ (exercise-induced)	+	+	+	[41,42,74,75]
miR-133a	−	−	−	−	−	[48,49,76,77]
miR-155	+	+	+	+ arteriogenic	+	[55,78,79]
miR-210	−	−	−	+	−	[58,80,81]

*AngII*, angiotensin II; TAC, transverse aortic constriction. + indicates that the miRNA promotes the function, − indicates that the miRNA inhibits the function.

**Table 3 diagnostics-11-01675-t003:** Comparison between miRNAs involved in fibrosis induced by MI vs. pressure overload.

Ref.	Signaling Pathway	Fibrosis Induced by MI	miRNA	Fibrosis Induced by Pressure Overload	Signaling Pathway	Ref.
[15]	*ARL2* *BCL2*	Pro-fibrosis	miR-15b	Anti-fibrosis (TAC)	*Smad3 TGFβRI*	[82]
[16]	*COL1α1, COL3α1, ELN, FBN, TGFβRII, PTEN*	Anti-fibrosis	miR-19a/b	No modification (TAC)		[16]
[16,18,19,20,21,22,23,24,25,26]	*Smad7* *Spry1* *ERK*	Pro-fibrosis	miR-21	Pro-fibrosis (TAC)	*Spry1 ERK*	[27]
[30]	*Smad1, BMP* *NF-κB* *COL1α1* *COL1α2*	Pro-fibrosis	miR-26a	Anti-fibrosis (TAC + *AngII*)	*CTGF* *NF-κB* *COL1α1*	[83]
[40]	*Smad4*	Pro-fibrosis	miR-34	Pro-fibrosis (TAC)	*VEGF* *POFUT1* *NOTCH1* *Vinculin* *Semaphorin 4B*	[84]
[56]	*NOTCH1* *JAG1* *DYRK1A*	Pro-fibrosis	miR-199b	Pro-fibrosis (TAC)	*DYRK1A*	[85]
[64,65]	*OCT4* *SOX2* TGFβRIII/p38	Pro-fibrosis	Let-7c Let-7i	Anti-fibrosis (*AngII*)	*IL-6* *COL1* *α* *2* *COL3* *α* *1* *COL4* *α* *1* *COL5* *α* *2*	[86]

*AngII*, angiotensin II; *ARL2*, ADP-Ribosylation Factor-Like Protein 2; *BCL2*, Apoptosis Regulator BCL2; *COL1α1*, collagen type 1, alpha 1 gene; *COL1α2,* collagen type 1, alpha 2 gene; *COL3α1*, collagen type 3, alpha 1 gene; *COL4α1,* collagen type 4, alpha 1 gene; *COL5α2,* collagen type 5, alpha 2 gene; *CTGF*, connective tissue growth factor; *DYRK1A*, dual specificity tyrosine-phosphorylation-regulated kinase 1A; *ELN*, elastin gene; *ERK*, extracellular signal-regulated kinase; *FBN*, fibrillin gene; *IL-6*, interleukin 6; *JAG1*, jagged 1 protein; *NOTCH1*, Notch receptor 1; *OCT4*, octamer-binding transcription factor 4; POFUT1, GDP-fucose protein O-fucosyltransferase 1; *PTEN*, phosphatase and tensin homolog protein; *Smad 1, 3, 4*, SMAD family members; TAC, transverse aortic constriction; *TGFβRI*, transforming growth factor-beta receptor I; *TGFβRII*, transforming growth factor-beta receptor II; *TGFβRIII*, transforming growth factor-beta receptor III; *VEGF*, Vascular endothelial growth factor.

**Table 4 diagnostics-11-01675-t004:** MiRNAs studied as prognostic biomarkers for LVR after MI.

miRNA	Effect on LVR	Study Population	Blood Sample Collection	Method for Assessing LVR	Definition of LVR	Reference
miR-1	+	80 patients with STEMI	On admission	CMR at 1 and 6 months	↑ >10% of ∆LVEDV	[87]
	−	44 patients with MI (SITAGRAMI trial)	d4, d9, 6 m	CMR at 4 d, 6 m	Absolute change for LVEDV	[88]
	No effect	12 patients with MI	d2, 5, 28, 90	Echo on d1, 5, 28, 90	Absolute change for LVEDV	[89]
miR-16/27a/101 /150	+ for miR-16/27a − for miR 150/101	150 patients with MI	At discharge	Echo at discharge and at a median of 176 d (range 128–262 d) after MI	Wall Motion Index Score > 1.2	[90]
miR-21	+	246 patients with anterior MI (REVE trial)	At discharge, 1 m, 3 m, 1 y	Echo on d3, 7, 3 m, 1 y	((EDV1year—EDVbaseline)/EDVbaseline)	[91]
	+	184 patients with MI	On admission	-	Death within 30 days	[92]
	+	198 patients with MI	d5	Echo on d5, 1 y	↑ >20% of LVEDV	[93]
	No effect	44 patients with MI (SITAGRAMI trial)	d4, d9, 6 m	CMR at 4 d, 6 m	Absolute change for LVEDV	[88]
	No effect	12 patients with MI	d2, 5, 28, 90	Echo on d1, 5, 28, 90	Absolute change for LVEDV	[89]
miR-29a	+	12 patients with MI	d2, 5, 28, 90	Echo on d1, 5, 28, 90	Absolute change for LVEDV	[89]
miR-29b	−	44 patients with MI (SITAGRAMI trial)	d4, d9, 6 m	CMR at 4 d, 6 m	Absolute change for LVEDV	[88]
miR-30a-5p	+	14 patients with STEMI	On admission, 6 m	Echo on admission, 6 m	LVEF ≤ 50% NT-proBNP ≥ 150pg/ml	[94]
miR-34a	+	359 patients with MI	On admission	Echo after admission, 6 m	↑ >10% of LVEDV	[95]
miR-92a	+	60 patients with MI	On admission and after PCI	Echo on admission, 3 m	Absolute change for LVEDV, LVESV, LVEF	[41]
miR-92	No effect	44 patients with MI (SITAGRAMI trial)	d4, d9, 6 m	CMR at 4 d, 6 m	Absolute change for LVEDV	[88]
miR-133a	No effect	246 patients with anterior MI (REVE trial)	At discharge, 1 m, 3 m, 1 y	Echo on d3, 7, 3 m, 1 y	((EDV1year—EDVbaseline)/EDVbaseline)	[96]
	No effect	12 patients with MI	d2, 5, 28, 90	Echo on d1, 5, 28, 90	Absolute change for LVEDV	[89]
miR-146a	+	198 patients with MI	d5	Echo on d5, 1 y	↑ >20% of LVEDV	[92]
miR-150	−	60 patients with MI	At discharge	Echo at discharge, 6 m	∆EDV > 0	[97]
miR-208b	+	359 patients with MI	On admission	Echo after admission, 6 m	↑ >10% of LVEDV	[95]
	+	100 patients with MI, 80 patients with UA 80 healthy controls	On admission and after PCI	Echo on admission, 6 m	↑ >10% of LVEDV	[98]
	+	21 patients with STEMI	On admission	-	Death within 6m	[99]
miR-208	No effect	12 patients with MI	d2, 5, 28, 90	Echo on d1, 5, 28, 90	Absolute change for LVEDV	[89]
miR-423-5p	No effect	246 patients with anterior MI (REVE trial)	At discharge, 1 m, 3 m, 1 y	Echo on d3, 7, 3 m, 1 y	((EDV1year—EDVbaseline)/EDVbaseline)	[96]
miR-1254	−	70 patients with STEMI	On admission	CMR at 1 w, 6 m	Absolute change for LVEDV, LVESV, LVEF	[100]

+ indicates that the miRNA promotes LVR,− indicates that the miRNA inhibits LVRCMR, cardiac magnetic resonance; d, day; Echo, echocardiography; EDV, end-diastolic volume; m, month(s); LVEDV, left ventricle end-diastolic volume; LVEF, left ventricle ejection fraction; LVESV, left ventricle end systolic volume; PCI, percutaneous coronary intervention; MI, myocardial infarction; NT-pro-BNP, pro-brain natriuretic peptide; STEMI, ST-elevation myocardial infarction; UA, unstable angina; y, year. Arrow means an increase.

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
