# Peer review of "MicroRNAs—The Heart of Post-Myocardial Infarction Remodeling"

_diagnostics, 2021, doi:10.3390/diagnostics11091675_

Round 1

Reviewer 1 Report

The Review Paper by Maries et al is a very well written review about the role of micro-RNAs in postinfarction remodeling. So far I did not really reflect this field, but this review is a very well introduction and comprehensive overview of knowledge in postmyocardial fibrosis regulated by micro-RNAs. To me, this is a sound review and it is publishable as is with a very few modifications/clarifications:

  1. Table 1: Effector cells: please explain the abbreviations CM, F, M and EC
  2. Line 228: "Therefore, an increase in blood miRNA-1 levels was ...." Were exactly in the blood were the miRNAs found, whole blood, serum, intracellular ? Please clarify .
  3. line 446: Thetherapeutic role of miRNAs...." . In the whole paper there was not much reference to the therapeutic use of miRNAs, e.g. to prevent fibrosis after myocardial infarction. The most parts of the paper are about pathyphysiology, fibrosis and prognosis. The authors may add a few lines to therapeutic interventions with references, if there are some available in literature. They mention in line 447, that there were some animal experiments. I would love to read more about these experiments, you may add a small chapter before the conclusion comes. 

Author Response

Dear Reviewer 1,

Thank you for critically evaluating our manuscript.

As requested, on line 228, we have replaced blood with plasma, thus refining the information regarding miR-1. Furthermore, we have introduced a new chapter (lines 420 – 476 and references 110-126) outlining the significant published results on the use of miRs/anti-miRs as therapeutic tools in myocardial infarction. Please see the attachment.

We would like to mention that the “CM,” “F,” “M,” and “EC” abbreviations are presented in the subtext of Table 1; we have outlined them in yellow for your convenience. 

Reviewer 2 Report

Presented review article tackles an interesting and important topic. Manuscript text, figures and tables are informative and well prepared. I have no comments.

Author Response

Thank you for critically evaluating our manuscript!